# High-Pressure Depolymerization of Poly(lactic acid) (PLA) and Poly(3-hydroxybutyrate) (PHB) Using Bio-Based Solvents: A Way to Produce Alkyl Esters Which Can Be Modified to Polymerizable Monomers

**DOI:** 10.3390/polym14235236

**Published:** 2022-12-01

**Authors:** Vojtěch Jašek, Jan Fučík, Lucia Ivanová, Dominik Veselý, Silvestr Figalla, Ludmila Mravcova, Petr Sedlacek, Jozef Krajčovič, Radek Přikryl

**Affiliations:** 1Institute of Materials Chemistry, Faculty of Chemistry, Brno University of Technology, 61200 Brno, Czech Republic; 2Institute of Environmental Chemistry, Faculty of Chemistry, Brno University of Technology, 61200 Brno, Czech Republic; 3Institute of Physical and Applied Chemistry, Faculty of Chemistry, Brno University of Technology, 61200 Brno, Czech Republic

**Keywords:** poly(lactic acid), poly(3-hydroxybutyrate), depolymerization, alcoholysis, methacrylation, polymerizable monomers, kinetics

## Abstract

The polyesters poly(lactic acid) (PLA) and poly(3-hydroxybutyrate) (PHB) used in various applications such as food packaging or 3D printing were depolymerized by biobased aliphatic alcohols—methanol and ethanol with the presence of *para*-toluenesulphonic acid (*p*-TSA) as a catalyst at a temperature of 151 °C. It was found that the fastest depolymerization is reached using methanol as anucleophile for the reaction with PLA, resulting in the value of reaction rate constant (*k*) of 0.0425 min^−1^ and the yield of methyl lactate of 93.8% after 120 min. On the other hand, the value of constant *k* for the depolymerization of PHB in the presence of ethanol reached 0.0064 min^−1^ and the yield of ethyl 3-hydroxybutyrate was of 76.0% after 240 min. A kinetics study of depolymerization was performed via LC–MS analysis of alkyl esters of lactic acid and 3-hydroxybutanoic acid. The structure confirmation of the products was performed via FT-IR, MS, ^1^H NMR, and ^13^C NMR. Synthesized alkyl lactates and 3-hydroxybutyrates were modified into polymerizable molecules using methacrylic anhydride as a reactant and potassium 2-ethylhexanoate as a catalyst at a temperature of 80 °C. All alkyl esters were methacrylated for 24 h, guaranteeing the quantitative yield (which in all cases reached values equal to or of more than 98%). The methacrylation rate constants (*k*′) were calculated to compare the reaction kinetics of each alkyl ester. It was found that lactates reach afaster rate of reaction than 3-hydroxybutyrates. The value of *k*′ for themethacrylated methyl lactate reached 0.0885 dm^3^/(mol·min). Opposite to this result, methacrylated ethyl 3-hydroxybutyrate’s constant *k*′ was 0.0075 dm^3^/(mol·min). The reaction rate study was conducted by the GC-FID method and the structures were confirmed via FT-IR, MS, ^1^H NMR, and ^13^C NMR.

## 1. Introduction

In recent years, processes producing plastic materials mostly use fossil-based polymers due to the fact that these molecules are cheap to obtain and their properties can be determined efficiently. However, this type of manufacturing is dependent on non-renewable resources which might lead to potential supply risks [1,2,3,4]. Therefore, various biopolymers based on bio-source inputs such as PLA or PHB attract a significant amount of attention [5,6]. Poly(lactic acid) (PLA) is a particular biopolymer used for numerous applications such as food packaging or 3D filament printing [2,7,8]. The increasing usage of this polymer leads to considerable problems—the rate of degradation of PLA in moderate environmental conditions is very slow; therefore, the accumulation of waste can occur [1,9,10]. There are ways to handle PLA waste, such as composting, incinerating, or mechanical recycling [7,11,12]. The first two processes do not generate any usable material and the mechanical recycling of PLA results in the production of a polymer with considerably worse properties than the original one. The reason for this outcome is that the thermal and photochemical degradation takes place during the mechanical processing of the PLA polymer, which leads to the decrease in molecular weight of the material [2,7,13,14]. Poly(3-hydroxybutyrate) (PHB) is a biopolymer that is produced by various microorganisms and can be degraded by living systems as well. The biodegradability of PHB is aconsiderable benefit of this polyester [2,15,16]. Nevertheless, PHB suffers rapid thermal degradation during the potential mechanical recycling, similarly to PLA, particularly at temperatures starting at 170–180 °C [2,17]. These two biopolymers are miscible, which is beneficial for a potential 3D printing usage, for example [18,19]. However, the fact that mixtures of numerous polymers and plasticizers are used complicates their potential recycling [20].

The chemical recycling of polyesters such as PLA or PHB is another way of handling leftover polymers or those with bad quality or properties such as low values of MW. This type of process results in forming monomers out of the polyester polymer chain. The produced monomers can either be carboxylic acids or esters depending on the chosen nucleophile. When water is used as a nucleophile, hydrolysis takes the place of the depolymerization process. Alcoholysis, on the other hand, requires alcohol to undergo the reaction, resulting in the formation of an ester via the transesterification mechanism [21,22,23,24,25,26,27]. These processes involve either specific reaction conditions (high pressure, high temperature) or the presence of a particular catalyst and appropriate organic solvents. Extreme conditions such as high temperature (above 120 °C) and high pressure (depending on the chosen nucleophile) can be complicated in terms of up-scaling or energy consumption [28,29,30,31]. On the other hand, specific catalysts can be expensive, and particularly organic solvents may be inappropriate for regeneration in the production process and lower temperatures could result in a decrease in the reaction rate [30,32]. The hydrolysis of PLA was experimentally verified using a temperature of 250 °C and high pressure. The molar ratio of 1:20 (PLA:H_2_O) and a reaction time of 10–20 min resulted in a 90% conversion to L-lactic acid [33,34,35]. Adding microwave irradiation to the process of hydrolysis with aratio of 1:3 (PLA:H_2_O) provided a 45% conversion to l-lactic acid after 120 min [34,35]. PHB alcoholysis was observed using ionic liquids as catalysts and methanol as a nucleophile. The highest yield of methyl 3-hydroxybutyrate (83.75%) was reached with the mixture of the molar ratio 5:1 (MeOH:PHB) and of the mass ratio 1:0.03 (PHB:cat.) with the conditions of a temperature of 140 °C and a reaction time of 3 h [36,37,38].

The main aim of this work is to describe a depolymerization process of the polyesters PLA and PHB in high-pressure and high-temperature conditions. The nucleophiles chosen for the reactions are methanol and ethanol and the reaction takes place in ahigh-pressure reactor. *Para*-toluensulphonic acid serves as an acidic catalyst for all alcoholyses. The monoester product structures are verified via numerous analyses (MS, FT-IR, ^1^H NMR, ^13^C NMR) and their reaction kinetics are studied. All synthesized monoesters (lactates and 3-hydroxybutyrates) undergo a methacrylation process to produce methacrylated esters which can be polymerized. The methacrylation reaction uses methacrylic anhydride as areagent and the chosen catalyst for this reaction is potassium 2-ethylhexanoate. The polymerizable products’ structures are verified by numerous methods (MS, FT-IR, ^1^H NMR, ^13^C NMR) as well.

## 2. Materials and Methods

### 2.1. Materials

PLA granulate was supplied from Fillamentum Manufacturing Czech s.r.o., Hulín, Czech Republic. Measured polymer parameters were as follows: number-average molecular weight (M_n_), 123,500 g/mol; weight-average molecular weight (MW), 235,300 g/mol; dispersity, 1.90. PHB powder was acquired from NAFIGATE Corporation a.s., Prague, Czech Republic. Measured polymer parameters were as follows: number-average molecular weight (M_n_), 85,040 g/mol; weight-average molecular weight (MW), 211,400 g/mol; dispersity, 2.49. All measurements of polymer properties were measured via GPC (Agilent 1100, Santa Clara, CA, USA) in chloroform (CHCl_3_) and the analysis parameters were as follows: mobile phase flow 1 mL/min; column temperature 30 °C, used column: PLgel 5 μm MIXED-C (300 × 7.5 mm). Aliphatic alcohols for depolymerization (methanol 99%, ethanol 99%) were supplied by Honeywell Research Chemicals, Charlotte, NC, USA (used alcohols were not claimed either synthetic or bio-source by the supplier). The catalyst for alcoholyses (*p*-toluensulphonic acid monohydrate), methacrylic anhydride (94%), potassium hydroxide (p.a.), d-chloroform (CDCl_3_; 99.8%), and 2-ethylhexanoic acid (for synthesis) were all acquired from Sigma-Aldrich, Prague, Czech Republic.

### 2.2. Methods for the Characterization of Products

#### 2.2.1. Fourier-Transform Infrared Spectrometry (FT-IR)

Infrared spectrometry was used as one of the structure verification methods, but itwas mainly supposed to serve as a confirmation of –OH hydroxyl functional groups in alkyl esters of either lactic acid or 3-hydroxybutanoic acid. Analyses were performed on the infrared spectrometer Bruker Tensor 27 (Billerica, MA, USA) by the attenuated total reflectance (ATR) method using diamond as a dispersion component. The irradiation source in this type of spectrometer is a diode laser. Due to the fact that instrumentation uses Fourier transformation, the Michelson interferometer was used for the quantification of the signal. Spectra were composed out of 32 total scans with a measurement resolution of 2 cm^−1^.

#### 2.2.2. Mass Spectrometry (MS)

MS conditions were as follows: ESI in positive mode; spray voltage: 3500 V; cone temperature 350 °C; cone gas flow: 35 a.u.; heated probe temperature: 650 °C; probe gas flow: 40 a.u., nebulizer gas flow: 55 a.u., and exhaust gas: ON. For quantification, MRM mode was used with the following MRM transitions for MeLa (RT 1.24 min; 105.1 > 84.6 with CE 0.25 eV;105.1 > 93.9 with CE 0.25 eV and 105.1 > 45.0 with CE 2 eV), for EtLa (RT 2.35 min; 119.1 > 47.3 with CE 2.0 eV and 119.1 > 91.1 with CE 2.0 eV) for M3HB (RT 1.91 min; 119.1 > 59.0 with CE 10 eV; 119.1 > 87.2 with CE 2.5 eV; and 119.1 > 101.2 with CE 1.0 eV), and for E3HB (RT 3.58 min; 133.1 > 73.3 CE with 5 eV and 133.1 > 87.2 with CE 5 eV). The collision gas Argon was used ata pressure of 1.5 mTorr.

Additionally, by the same MS method, newly synthesized monomers methacrylated methyl lactate (MeLaMMA), methacrylated ethyl lactate (EtLaMMA), methacrylated methyl 3-hydroxybutyrate (M3HBMMA), and methacrylated ethyl 3-hydroxybutyrate (E3HBMMA) were qualitatively characterized by product scan; therefore, themass spectra of these compounds were obtained. The precursor of MeLAaMMA (*m/z* 173.1) was fragmented (CE 10 eV) and the following product ions were obtained: 57.3 and 74.2. The precursor of EtLaMMA (*m/z* 187.0) was fragmented (CE 2.5 eV) and the following product ions were obtained: 69.2; 113.1 and 141.1. The precursor of M3HBMMA (*m/z* 187.1) was fragmented (CE 2.5 eV) and the following product ions were obtained: 59.3; 69.2; 101.1 and 155.1. The precursor of E3HBMMA (*m/z* 201.0) was fragmented (CE 2.5 eV) and the following product ions were obtained: 69.0; 73.3; 115.1; and 155.0.

#### 2.2.3. Nuclear Magnetic Resonance (NMR)

Nuclear magnetic resonance was used to obtain ^1^H and ^13^C spectra to confirm the structure of synthesized molecules. The measurements were conducted by instrument Bruker Avance III 500 MHz (Bruker, Billerica, MA, USA) with the measuring frequency of 500 MHz for ^1^H NMR and 126 MHz for ^13^C NMR at the temperature of 30 °C using d-chloroform (CDCl_3_) as a solvent with tetramethylsilane (TMS) as an internal standard. The chemical shifts (*δ*) are expressed in part per million (ppm) units which are referenced by a solvent. Coupling constant *J* has (Hz) unit with coupling expressed as s—singlet, d—doublet, t—triplet, q—quartet, p—quintet, m—multiplet.

### 2.3. Alcoholyses of Polyesters

All depolymerization reactions took place in the high-pressure reactor of a volume reservoir of 1.8 L. Mixtures consisting of the polyester (PLA or PHB) and the alcohol (methanol or ethanol) in amolar ratio 1:4 (PLA/PHB:MeOH/EtOH) with the presence of dissolved catalyst p-toluensulphonic acid in a molar ratio 1:0.01 (PLA/PHB:*p*-TSA) were transferred into the reactor (see Figure 1 for PLA and Figure 2 for PHB). The reactor was heated up to 151 °C and the pressure increased regarding the particular type of suspense according to the vapor-pressure characteristics of the alcohol and forming ester product (ranging from 7 to 15 bar). The samples for kinetics analysis were taken directly from the reactor and cooled down immediately. The kinetics of forming monoesters was monitored via LC–MS analysis. The conversion of the particular polyester was calculated from the leftover polymer acquired by mixing the reaction solution sample with water. The unreacted polymer was precipitated and weighed. After the alcoholyses were stopped (depending on the particular combination of reagents) the leftover alcohol and the formed monoester were distilled from the solution. Products of the depolymerization were analyzed via MS, FT-IR, ^1^H NMR, and^13^C NMR for structure verification. Yields of monoester (Yield) and polymer conversions (*X*) were calculated as follows:(1)$$X=\frac{\mathrm{Starting}\;\mathrm{polymer}\;\mathrm{weight}\;-\;\mathrm{leftover}\;\mathrm{polymer}\;\mathrm{weight}}{\mathrm{Starting}\;\mathrm{polymer}\;\mathrm{weight}}\;\times \;100\%$$
(2)$$\mathrm{Yield}=\frac{\mathrm{Measured}\;\mathrm{quantity}\;\mathrm{of}\;\mathrm{ester}}{\mathrm{Theoretical}\;\mathrm{quantity}\;\mathrm{of}\;\mathrm{ester}}\times \;100\%$$

**Methyl lactate (MeLa):**^1^H NMR (Appendix A) (CDCl_3_, 500 MHz): δ(ppm) 4.30–4.26 (q; *J* = 6.9 Hz; 1H), 3.78 (s; 3H), 2.80 (s; 1H), 1.42–1.41 (d; *J* = 6.9 Hz; 3H). ^13^C NMR (Appendix A) (CDCl_3_, 126 MHz): *δ*(ppm) 176.26; 66.90; 52.65; 20.50.

**Ethyl lactate (EtLa):**^1^H NMR (Appendix A) (CDCl_3_, 500 MHz): δ (ppm) 4.27–4.21 (m; 3H), 2.84 (s; 1H), 1.41–1.40 (d; *J* = 6.9 Hz; 3H), 1.31–1.28 (t; *J* = 7.16; 7.16 Hz; 3H). ^13^C NMR (Appendix A) (CDCl_3_, 126 MHz): δ (ppm) 175.72; 66.76; 61.64; 20.38; 14.15.

**Methyl 3-hydroxybutyrate (M3HB):**^1^H NMR (Appendix A) (CDCl_3_, 500 MHz): δ (ppm) 4.22–4.16 (qd; *J* = 8.54; 6.30; 6.27; 6.27 Hz; 1H), 3.70 (s; 3H), 2.93 (s; 1H), 2.51–2.40 (m; 2H), 1.23–1.22 (d; *J* = 6.30 Hz 3H). ^13^C NMR (Appendix A) (CDCl_3_, 126 MHz): δ (ppm) 173.29; 64.28; 51.71; 42.63; 22.49.

**Ethyl 3-hydroxybutyrate (E3HB):**^1^H NMR (Appendix A) (CDCl_3_, 500 MHz): δ 4.22–4.15 (q; *J* = 7.16; 7.12; 7.12 Hz; 3H), 2.98 (s; 1H), 2.50–2.46 (dd; *J* = 16.38; 3.48 Hz; 1H), 2.43–2.38 (dd; *J* = 16.40; 8.68 Hz; 1H), 1.28–1.25 (t; *J* = 6.52; 6.52 Hz; 3H), 1.23–1.22 (d; *J* = 6.93 Hz; 3H). ^13^CNMR (Appendix A) (CDCl_3_, 126 MHz): δ (ppm) 172.93; 64.30; 60.67; 42.83; 22.45, 14.19.

### 2.4. Methacrylation of Alkyl Esters

Synthesized alkyl esters of either lactic acid or 3-hydroxybutanoic acid did undergo a reaction with methacrylic anhydride (MAA) in order to form a polymerizable monomer. The reaction mixtures were prepared in a molar ratio 1:1 (ester:MAA) (see Figure 3 for lactates and Figure 4 for 3-hydroxybutyrates). The reaction mixture was poured into a three-necked round bottom flask and placed in an oil bath tempered at 80 °C and stirred via a magnetic stirrer. The reactions were catalyzed by a 50% solution of potassium 2-ethylhexanoate in 2-ethylhexynoic acid (2-EHA) that was prepared via a neutralization reaction of 2-ethylhexanoic acid with potassium hydroxide in a mass ratio 1:2 (KOH:acid) while the reaction water was evaporated. The catalyst was added to the mixture in a molar ratio 1:0.02 (ester:catalyst). The reaction started the moment the catalyst was added and all reactions took 24 h of reaction time. The conversions of reactants and the yields of forming products were monitored via GC-FID analysis. The formed methacrylic acid was neutralized by potassium hydroxide aqueous solution and separated from the metracrylated product. Methacrylated alkyl ester structures were verified via MS, FT-IR, ^1^H NMR, and^13^C NMR methods.

**Methacrylated methyl lactate (MeLaMMA):**^1^H NMR (Appendix A) (CDCl_3_, 500 MHz): δ(ppm) 6.20–6.19 (dd; *J* = 1.54; 0.94 Hz; 1H), 5.63–5.62 (p; *J* = 1.53; 1.53; 1.52; 1.52 Hz; 1H), 5.17–5.13 (q; *J* = 7.08; 7.07; 7.07 Hz; 1H), 3.75 (s; 3H), 1.97–1.96 (dd; *J* = 1.6; 1.00 Hz; 3H), 1.53(d; *J* = 7.00 Hz; 3H). ^13^C NMR (Appendix A) (CDCl_3_, 126 MHz): δ (ppm)171.30; 166.68; 135.63; 126.44; 68.77; 52.27; 18.13; 16.96.

**Methacrylated ethyl lactate (EtLaMMA):**^1^H NMR (Appendix A) (CDCl_3_, 500 MHz): δ(ppm) 6.20–6.19 (p; *J* = 1.07; 1.07; 1.07; 1.07 Hz; 1H), 5.63–5.61 (p; *J* = 1.57; 1.57; 1.57; 1.57Hz; 1H), 5.14–5.10 (q; *J* = 7.07; 7.07; 7.03 Hz; 1H), 4.23–4.18 (q; *J* = 7.16; 7.16; 7.15 Hz; 2H), 1.97–1.96 (dd; *J* = 1.58; 1.01 Hz; 3H), 1.53–1.52 (d; *J* = 7.05 Hz; 3H), 1.28–1.26 (t; *J* = 7.15; 7.15 Hz; 3H); 1.53–1.52 (d; *J* = 7.05 Hz; 3H), 1.28–1.26 (t; *J* = 7.15; 7.15 Hz; 3H). ^13^C NMR (Appendix A) (CDCl_3_, 126 MHz): δ (ppm) 170.86; 166.76; 135.74; 126.35; 68.93; 61.31; 18.18; 16.97; 14.11.

**Methacrylated methyl 3-hydroxybutyrate (M3HBMMA):**^1^H NMR (Appendix A) (CDCl_3_, 500 MHz): δ (ppm) 6.07–6.06 (dq; *J* = 1.96; 1.02; 0.98; 0.98 Hz; 1H), 5.56–5.53 (p; *J* = 1.60; 1.60; 1.58; 1.58 Hz; 1H), 5.35–5.29 (dp; *J* = 7.32; 6.26; 6.26; 6.25; 6.25 Hz; 1H), 3.68 (s; 3H), 2.72–2.67 (dd; *J* = 15.34; 7.29 Hz; 1H), 2.57–2.53 (dd; *J* = 15.35; 5.79 Hz; 1H), 1.92 (dd; *J* = 1.63; 1.01 Hz; 3H), 1.35–1.34 (d; *J* = 6.36 Hz; 3H). ^13^C NMR (Appendix A) (CDCl_3_, 126MHz): δ (ppm) 170.70; 166.59; 136.47; 125.41; 67.68; 51.73; 40.74; 19.89; 18.23.

**Methacrylated ethyl 3-hydroxybutyrate (E3HBMMA):**^1^HNMR (Appendix A) (CDCl_3_, 500 MHz): δ (ppm) 6.07–6.06 (dd; *J* = 1.75; 0.97 Hz; 1H), 5.54–5.53 (q; *J* = 1.63; 1.63; 1.63 Hz; 1H), 5.35–5.29 (dp; *J* = 7.50; 6.24; 6.24; 6.24; 6.24 Hz; 1H), 4.16–4.10 (qd; *J* = 7.11; 7.06; 7.06; 0.96Hz; 2H), 2.69–2.65 (dd; *J* = 15.28; 7.42 Hz; 1H), 2.55–2.51 (dd; *J* = 15.29; 5.75 Hz; 1H), 1.94–1.91 (m; 3H), 1.34–1.33 (d; *J* = 6.28 Hz; 3H), 1.25–1.22 (t; *J* = 7.13; 7.13 Hz; 3H). ^13^C NMR (Appendix A) (CDCl_3_, 126 MHz): δ (ppm) 170.21; 166.54; 136.46; 125.33; 67.71; 60.57; 41.01; 19.85; 18.19; 14.14.

### 2.5. Methods for the Reaction Kinetics Study

#### 2.5.1. LC–MS Method for Depolymerization Kinetics

Samples were obtained from the reactor during organic synthesis, followed by quantification of products methyl lactate (MeLa), ethyl lactate(EtLa), methyl 3-hydroxybutyrate(M3HB), and ethyl 3-hydroxybutyrate(E3HB) by ultra-performance liquid chromatography (UHPLC Agilent 1290 Infinity LC) in tandem with triple quadruple (Bruker EVOQ LC-TQ) (Billerica, MA, USA) with atmospheric pressure electrospray ionization (ESI). An external generator of gases was used as the source of nitrogen and air (Peak Scientific—Genius 3045) (Inchinnan, UK). As a stationary phase column, Luna^®^ Omega Polar C18 Phenomenex (100 × 2.1 mm, 1.6 µm) was used. The optimum column temperature was adjusted to 40 °C and the flow rate was set to 0.5 mL/min. The mobile phases were as follows: (A) 0.1% HCOOH in H_2_O and (B) ACN were used with the following gradient program of An eluent (%): t (0 min) = 90, t (0.5 min) = 85, t (3.5 min) = 5, t (4.5 min) = 95. Stop time was set to 6.0 min and re-equilibration time was set to 2.0 min. The injection volume applied in all analyses was 7 µL.

MS conditions were as follows: ESI in positive mode; spray voltage: 3500 V; cone temperature 350 °C; cone gas flow: 35 a.u.; heated probe temperature: 650 °C; probe gas flow: 40 a.u., nebulizer gas flow: 55a.u. and exhaust gas: ON. For quantification, MRM mode was used with the following MRM transitions for MeLa (RT 1.24 min; 105.1 > 84.6 with CE 0.25 eV; 105.1 > 93.9 with CE 0.25 eV and 105.1 > 45.0 with CE 2 eV), for EtLa (RT 2.35 min; 119.1 > 47.3 with CE 2.0 eV and 119.1 > 91.1 with CE 2.0 eV), for M3HB (RT 1.91 min; 119.1 > 59.0 with CE 10eV; 119.1 > 87.2 with CE 2.5 eV and 119.1 > 101.2 with CE 1.0 eV), and for E3HB (RT 3.58 min; 133.1 > 73.3 CE with 5eV and 133.1 > 87.2 with CE 5 eV). The collision gas Argon was used ata pressure of 1.5 mTorr.

#### 2.5.2. GC-FID Method for Methacrylation Kinetics

Samples were obtained from the reactor during organic synthesis, followed by quantification of reactants (alkyl esters, methacrylic anhydride) by gas chromatography (Hewlett Packard 5890 Series II) (Palo Alto, CA, USA) with a flame ionization detector (FID). Gas bottles of nitrogen (as auxiliary gas for FID), air (as an oxidizer for FID), and hydrogen (as carrier gas and fuel for FID) were used. Capillary GC column ZB-624 (60 m × 0.32 mm, 1.8 µm) served as a stationary phase. The temperature of the inlet was set to 200 °C and the temperature of the detector to 260 °C. Substances were separated with atemperature gradient, with an initial temperature of 60 °C (held for 1 min) followed by atemperature rate of 20 °C/min with a final temperature of 250 °C (held for 15 min). The column flow rate set for the analyses was 3 mL/min and the split ratio was 1:40. The injection volume applied in all analyses was 1 µL. The retention time of peaks: MeLa (RT6.15 min); EtLa (RT 7.05 min); M3HB (RT 7.76 min); E3HB (RT 8.53 min); MAA (RT 9.88 min); MeLaMMA (RT 9.80); EtLaMMA (RT10.45 min); M3HBMMA (RT 10.85); E3HBMMA (RT 14.48 min).

## 3. Results

### 3.1. Depolymerization of PLA and PHB via Alcoholysis

The reaction mixtures for the depolymerization were prepared according to the mass proportion shown in Table 1. The mass of a particular polymer in every mixture was constant and the amount of reacting nucleophile (alcohol) changed depending on the molar ratio of the treactants. The amounts of catalyst (*p*-TSA) for each reaction solution are written in the table as well. Table 1 also contains information on the boiling points of each alkyl ester that was synthesized.

The results of the LC–MS analysis in Figure 1a show that methyl esters of each lactic or 3-hydroxybutanoic acid reach their reaction equilibria after about 90 min of depolymerization (MeLawas slightly faster than M3HB), resulting in yields of 93.8% for MeLa and 91.6% for M3HB. On the other hand, the ethyl esters of both carboxylic acids did not reach their total yield value after 4 h of reaction. The reaction rate of ethyl 3-hydroxybutyrate seems to be the slowest, reaching a yield of 76.0% after 240 min. The ethyl lactate’s yield after the same reaction time reached 85.1%. The differences between the product yields could be caused by the steric effects of the particular molecules involved in the reaction.

The conversion signs of progress of either PLA or PHB during the depolymerization reactions are displayed in Figure 1b. They have been acquired by weighing the residual polymer from the taken sample. The precipitated content of the unreacted polymer was measured and the percentage of conversion was calculated. The results have very similar data curves as the values of the products’ yields due to the fact that these values are connected. The conversion of PLA is 94.8% for methanolysis (after 120 min) and the conversion of PLA is 89.1% for ethanolysis (after 240 min). PHB reached conversions values of 92.8% (MeOH, 120 min) and 80.0% (EtOH, 240 min).

The pressure of the reacting mixture was monitored during every alcoholysis. The results shown in Figure 2 confirm the progressing depolymerization for each mixture. Due to the incorporation of alcohol into the structure of the alkyl ester, the pressure in the system should decrease as a result of decreasing the presence of evaporating alcohol. These expectations are fulfilled except for the methyl lactate. The pressure of the system for the methanolysis of PLA increases after 45 min of reaction. This elevation is caused by the forming MeLa since the reaction temperature was 151 °C and the boiling point value of MeLa is below this temperature (shown in Table 1). Therefore, the occurring monoester participates in the pressure increase.

### 3.2. Kinetics of the Depolymerization of PLA and PHB via Alcoholysis

The first-order reaction order is mostly used to describe the alcoholysis depolymerization due to the fact that if the excess of alcohol is added to the mixture, the concentration of polymer molecules affects the reaction kinetics directly due to its lesser molar amount of mixture [43,44]. If the first-order reaction is used for the calculation, the equations are the following:(3)$$r=\frac{dc_{polymer}}{dt}=-kc_{polymer},$$
where *r* is the reaction rate (mol/(dm^3^·min)), *k* represents the reaction constant (min^−1^), and *c_polymer_* stands for the molar concentration of the polymer (mol/dm^3^) at time *t* (min). The molar concentration of the polymer as a reagent can be substituted by the following conversion values:(4)$$c_{polymer}=\left(1-X\right),$$
where the molar concentration (*c_polymer_*) is expressed by conversion (*X*). This equation is applied for the values of conversion from the number interval of <0,1>. If the molar concentration is replaced with the conversion, the first-order equation has to be rewritten:(5)$$\frac{dX}{dt}=k\left(1-X\right)$$

If Equation (5) is calculated generally, the steps are as follows:(6)$$\frac{dX}{\left(1-X\right)}=kdt$$
(7)$$\int_{0}^{X}\frac{dX}{\left(1-X\right)}=k\int_{0}^{t}dt$$
(8)−[ln(1−X)−ln(1)]=kt
(9)$$\ln\frac{1-X}{1}=-kt$$
(10)$$\ln\frac{1}{1-X}=kt$$

Equation (10) was applied to the kinetics values measured by LC–MS analysis. The conversion was transferred into a modified form using the logarithm of a fraction and the dependence on time, as shown in Figure 3. There is evidence that the highest reaction rate constant comprises the depolymerization of PLA in methanol (producing MeLa) with the value of approximately 0.0425 min^−1^. On the other hand, the depolymerization of PHB in ethanol provided the lowest value of reaction rate constant, reaching approx. 0.0064 min^−1^. The rest of the reaction rate constants (*k*) are shown in Table 2.

### 3.3. Structural Characterization of Synthesized Alkyl Esters

#### 3.3.1. FT-IR Analyses of Alkyl Esters

Infrared spectrometry using Fourier transformation was used as the first method to confirm the distilled synthesized product. The main peaks shown in Figure 4 lay in the intervals of wave numbers of either 3700–3200 cm^−1^ (–OH stretching) or the values of 1210–1163 cm^−1^ (C=O stretching) and 1750–1735 cm^−1^ (C–O stretching). In particular, the ester bond stretching signals are important since the initial suspension mixture would contain aliphatic alcohol, which provides the signal for the hydroxyl functional groups as well.

#### 3.3.2. MS Analyses of Alkyl Esters

The analytic procedure to obtain all MS spectra of synthesized alkyl esters of lactic and 3-hydroxybutyric acid is described above (see Section 2.2.2). All molecular precursor ions have been measured as products of the ESI ionization process. Other signals are particular product ions occurring due to the fragmentation of the molecules apart during the MS/MS analysis. All spectra are shown in Figure 5.

### 3.4. Methacrylation of the Alkyl Esters of Lactic and 3-Hydroxybutanoic Acid

The reaction mixtures for the syntheses of methacrylated alkyl esters of carboxylic acids were prepared according to the mass proportion shown in Table 3. The mass of a particular alkyl ester in every mixture was constant and the amount of reacting methacrylic anhydride changed depending on the molar ratio of the reactants. The amount of catalyst (50% solution of potassium 2-ethylhexanoate in 2-ethylhexanoic acid) was calculated according to the particular reacting alkyl ester.

The results of the methacrylation reaction that forms methacryled alkyl esters of lactic or 3-hydroxybutanoic acid are shown in Figure 6. Both reactants which were monitored (alkyl ester and methacrylic anhydride) via GC-FID analysis have similar time progressions of their conversion values due to the fact that their molar ratio in the reacting mixtures was 1:1 mol in all cases. The conversion values of methacrylic anhydride are slightly higher, likely due to the fact that the anhydride participated in secondary reactions in the reaction mixture (water hydrolysis, etc.). It is evident that the esters of lactic acid (MeLa and EtLa), formed into the products MeLaMMA and EtLaMMA, respectively, progressed faster in time than the alkyl esters of 3-hydroxybutanoic acid. These results may have been determined by the steric effects of each ester and due to their varying polarity which could have affected the effectiveness of the catalyst.

The methacrylation reactions were performed for 24 h to obtain the highest yields of the methacrylated product. The graphs below are presented in order to compare the rates of each particular reaction which served for the calculation of reaction rate constants. The yields of methacrylated alkyl esters products increasing in time are shown in Table 4. The yield quantified after 5 h of reaction is shown for comparison.

### 3.5. Kinetics of the Methacrylation Reactions

It is assumed that when equimolar amounts of both reactants (alkyl ester and methacrylic anhydride) are used, the rate of methacrylation is dependent on the concentration of both reactants. The acylation of hydroxyl functional groups using homogeneous catalysis was considered as potential reaction kinetics (methacrylation reaction is a type of the acylation) [45]. The Equations defining the dependence of the concentration of both reactants (alkyl ester and methacrylic anhydride) on time leading to acquiring the reaction rate constant are as follows:(11)$$r^{\prime }=\frac{dc_{ester}}{dt}=\frac{dc_{MAA}}{dt}=-k^{\prime }c_{ester}c_{MAA},$$

Equation (11) defines the conventional second-order rate where *r’* is the reaction rate (mol/(dm^3^·min)), *k*′ represents the reaction constant (dm^3^/(mol·min)), *c_ester_* stands for the molar concentration of a particular alkyl ester (mol/dm^3^), and *c_MAA_* stands for the molar concentration of methacrylic anhydride (mol/dm^3^) at time *t* (min). To solve the second-order rate of reaction, several mathematical adjustments need to be made:(12)$$c^{0}{}_{ester}=a$$
(13)$$c^{0}{}_{MAA}=b$$
(14)$$c_{ester}=a-x$$
(15)$$c_{MAA}=b-x,$$
where *x* stands for the concentration of each reactant in particular time *t* (min), *c*^0^*_ester_* (mol/dm^3^) is the initial concentration of alkyl ester, and *c*^0^*_MAA_* (mol/dm^3^) is the initial concentration of methacrylic anhydride. Considering these additional defined quantities, Equation (11) can be rearranged and solved:(16)$$-\frac{dx}{dt}=-k^{\prime }\left(c^{0}{}_{ester}-x\right)\left(c^{0}{}_{MAA}-x\right)$$
(17)$$\int_{0}^{x}\frac{dx}{\left(c^{0}{}_{ester}-x\right)\left(c^{0}{}_{MAA}-x\right)}=k^{\prime }\int_{0}^{t}dt$$
(18)$$\frac{1}{b-a}(\ln\frac{1}{a-x}-\ln\frac{1}{b-x})=k^{\prime }t$$
(19)$$\frac{1}{c^{0}{}_{MAA}-c^{0}{}_{ester}}(\ln\frac{c^{0}{}_{ester}}{c^{0}{}_{ester}-x}-\ln\frac{c^{0}{}_{MAA}}{c^{0}{}_{MAA}-x})=k^{\prime }t,$$

The left side of Equation (19) can be simplified by applying the rule of logarithm, and when the simplified equation has been rearranged, the dependence of the actual concentration during the reaction on time can be formed:(20)$$\frac{1}{c^{0}{}_{MAA}-c^{0}{}_{ester}}\ln\frac{c^{0}{}_{ester}c_{MAA}}{c^{0}{}_{MAA}c_{ester}}=k^{\prime }t$$
(21)$$\ln\frac{c^{0}{}_{ester}c_{MAA}}{c^{0}{}_{MAA}c_{ester}}=k^{\prime }\left(c^{0}{}_{MAA}-c^{0}{}_{ester}\right)t$$

Equation (21) can be used to obtain the reaction rate constant *k*′. If the graphic solution is applied, the slope of the linear curve acquired from the graph contains the constant *k*′. All data gathered during the reaction progress in time recalculated for mathematical purposes are shown in Figure 7. All reaction rate constants are written in Table 5. It is evident from the results that, in general, the methacrylation reactions of alkyl esters of lactic acid have higher reaction rate constants than the alkyl esters of 3-hydroxybutanoic acid. It is assumed that the availability of the hydroxyl functional group of lactates is better than for 3-hydroxybutyrates.

### 3.6. Structural Characterization of Synthesized Methycrylated Alkyl Esters

#### 3.6.1. FT-IR Analyses of Methacrylated Alkyl Esters

Infrared spectrometry using Fourier transformation’s results for the confirmation of the structures of the synthesized methacrylated alkyl esters are displayed in Figure 8. Peaks showing the presence of signals, which belong to ester bonds, lay in the intervals of wave numbers of either 1210–1163 cm^−1^ (C=O stretching) or the values of 1750–1735 cm^−1^ (C–O stretching). The signals referring to C–O stretching are split in every spectrum. The reason for the splitting of the peak is the presence of two different types of ester bonding in molecules. One bond belongs to the ester of lactic or 3-hydroxybutanoic acid and aliphatic alcohol. The other signal refers to the ester bond between the formed alkyl ester of carboxylic acid and the methacrylic acid. The second type of signal that can be found in FT-IR spectra reaches the values of wave numbers of either 1670–1600 cm^−1^ (C=C stretching) or 1000–650 cm^−1^ (C=C bending). These peaks uncover the presence of unsaturated double bonds within the structures of synthesized products that belong to methacrylates. Another confirmation of the successful methacrylation is the absence of signal in the area of 3700–3200 cm^−1^ (–OH stretching). These functional groups were supposed to react with methacrylic anhydride. Therefore, their peaks are missing in comparison with FT-IR spectra in Figure 4.

#### 3.6.2. MS Analyses of Methacrylated Alkyl Esters

The analytic procedure to obtain all MS spectra of synthesized methacrylated alkyl esters is described above (see Section 2.2.2). All molecular precursor ions have been measured as products of the ESI ionization process. Other signals are particular product ions occurring due to the fragmentation of the molecules during the MS/MS analysis. All spectra are shown in Figure 9.

## 4. Discussion

This work was focused on the experimental confirmation of the depolymerization of poly(lactic acid) (PLA) and poly(3-hydroxybutyrate) (PHB) via alcoholysis. All reactions were performed in identical conditions, which were a temperature of 151 °C, the presence of the constant amount of molar alcohol access (4:1) and the catalyst *para*-toluenesulphonic acid, particularly 1% mol. of the particular polyester. The pressure of the reaction solution differed regarding the used alcohol and the particular reaction combination. In all cases, the pressure in the reactor decreased over time except for the mixture containing methanol and poly(lactic acid). This mixture’s pressure elevated from 12.54 bar to 13.40 due to the boiling point value of methyl lactate being 145 °C, which means this ester evaporated in the reactor as well. Generally, lower pressure values were measured for theethanolyses of both PLA and PHB, which decreased from an approximate value of 8.63 bar to 7.54 (E3HB) and 7.39 (M3HB). The decrease is the consequence of lowering the volatility of the reacting solution due to the forming of alkyl esters. It was also found that the rates of methanolyses are faster than the rates of ethanolyses. Depolymerization reaction rate constants (*k*) were calculated for all experimental reactions, resulting in the highest one of 0.0425 (min^−1^) for the methanolysis of PLA and the lowest constant related to ethanolysis of PHB, reaching a value of 0.0064 (min^−1^). These differences are probably caused by the steric effects of particular reactants. The yields of each product were: 93.79% MeLa (120 min); 91.64% M3HB (120 min); 85.08% EtLa (240 min); and 76.03% E3HB (240 min).

The synthesized alkyl esters from the depolymerization of PLA and PHB were modified by the methacrylation reaction with methacrylic anhydride (MAA) to methacrylated alkyl esters, which are polymerizable. The reaction mixtures were composed of equal mass amounts of alkyl esters and different amounts of methacrylic anhydride in equimolar ratio to the esters. The catalyst used for the methacrylation was potassium 2-ethylheaxoate (50% mass solution) in 2-ethylhexanoic acid. The amount of catalyst was 2% mol. The rates of the methacrylation at 80 °C of each alkyl ester define the calculated reaction rate constants (*k*′). It has been observed that in general, the methacrylation of alkyl lactates was faster than in the case of alkyl 3-hydroxybutyrates. The highest value of methacrylation rate constant comprises the mixture containing methyl lactate (*k*′ = 0.0885 dm^3^/(mol·min)).On the other hand, the lowest one belongs to ethyl 3-hydroxybutyrate (*k*′ = 0.0079 dm^3^/(mol·min)). These constants have been calculated according to the reactions’ progress in time to evaluate a comparison for all reactants, but the methacrylation reactions of all alkyl esters were performed for 24 h to ensure the biggest possible yield. All yields reached values equal to or higher than 98%.

## 5. Conclusions

Several conclusions from the performed experiments can be summarized. The type of aliphatic alcohol plays a major role in the rate of depolymerization of the polyesters PLA and PHB. Methanol ensures a faster depolymerization process than ethanol. However, using methanol can be problematic regarding the used equipment due to the fact that these mixtures produce a higher overpressure. The methacrylation of all alkyl esters of lactic or 3-hydroxybutanoic acid reaches high yields after 24 h, close to 100%. Nevertheless, the lactic esters undergo methacrylation reaction at a faster rate than the 3-hydroxybutyrates. The distillation of the forming methacrylic acid could be performed instead of neutralizing and washing the acid; however, the appropriate stabilization against spontaneous polymerization has to be ensured.

## Data Availability

Not applicable.

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
