# Peer review of "High-Pressure Depolymerization of Poly(lactic acid) (PLA) and Poly(3-hydroxybutyrate) (PHB) Using Bio-Based Solvents: A Way to Produce Alkyl Esters Which Can Be Modified to Polymerizable Monomers"

_polymers, 2022, doi:10.3390/polym14235236_

Round 1

Reviewer 1 Report

It is my pleasure to review the manuscript submitted by Jašek et al. In this article (polymers-2052066-v1) the authors conducted high-pressure depolymerization/alcoholysis of poly(lactic acid) (PLA) and poly(3-hydroxybutyrate) (PHB) in presence of a p-TSA catalyst. The monoesters thus produced were converted into corresponding methacrylates. In addition, a kinetics study of depolymerization was performed via LC-MS analysis of alkyl esters of lactic acid and 3-hydroxybutanoic acid.

The study is interesting. However, authors could have checked the alcoholysis process at low temperatures (or can conducted a comparison between low and high-temperature depolymerization). Once more, the manuscript needs structural modification (i. Spectral data should be mentioned in the experimental section and not with figures; ii. 1H and 13C NMR spectra should be added in a supplementary file).

Some of the flaws are mentioned below:

[1] Title: The title seems to be lengthy. However, can it be written without abbreviation such as – High-Pressure Depolymerization of Poly(Lactic Acid) and Poly(3-Hydroxybutyrate) Using Bio-based Solvents and Synthesis of Methacrylate Esters

[2] Abstract and keywords: Many typos must be corrected.

- Please write ‘poly(3-hydroxybutyrate)’instead of ‘poly(3hydroxybutyrate)’ throughout the manuscript

- Need to use ‘p-toluensulphonic acid’ or ‘para-toluensulphonic acid’ instead of ‘p-toluensulphonic acid’ or ‘para-toluensulphonic acid’.

[3] Introduction: Please check and improve lines 83-94 for better understanding.

 - Abbreviated form of molecular weight should be MW.

[4] Materials and methods: This section should be restructured as - 2.1. Materials; 2.2. Methods for the characterization of products (also make this section short); 2.3. Alcoholyses of polyesters; 2.4. Methacrylation of alkyl esters; 2.5. Methods for the reaction kinetics study.

- In subsection 2.3. Alcoholyses of polyesters, write the name of products and their spectral data. Similarly, in subsection 2.4. Methacrylation of alkyl esters, write the name of products and their spectral data (for more information, authors may check doi: 10.1016/j.carres.2020.108130; 10.1016/j.carres.2019.107812).

[5] Results: It is common in research articles that 1H and 13C NMR spectra are added to the supplementary files. Only significant spectra are shown in the main manuscript. Please check any good organic chemistry articles.

- In sections 3.3.3,3.3.4, 3.6.3, and 3.6.4, remove spectral data from all Figure titles (as these will be mentioned in the materials/experimental section (this is truly standard procedure).

- In line 310, is it Chapter or section?

- Please improve lines 431-432.

[6] Discussion: This part can be improved.

[7] Conclusions: Please change/improve the first sentence of this section. Also, write h (instead of hours, for example, 4 h, 5 h, 24 h, etc.) throughout the manuscript.

[8] Reference: References should be journal style.

Reviewer 2 Report

Generally this article is above average and highly detailed. The work is generally sounds and reproducible. The reviewer doesn't have much comments, only with several opinions below:

1. Seems like your methanol/ethanol acquired from Honeywell directly. Are they claimed as bio-based? since your title is "bio-based solvents". If not, better clarified this in the experimental section, which is a proof-of-concept substitution.

2. PDI, or polydispersity index is a depreciated term by IUPAC, all that appeared in this article must replaced by "dispersity" instead.

3. Is there any empty set of reactions without using any alcohol? as baseline comparison. Also, what is the authors opinion of residual catalyst (i.e., Tin 2-Ethylhexanoate, or Tin Octoate) for the synthesis of PLA and PHB ? In other word, residual catalyst for "polymerization" may (or may not) influence "depolymerization" in your current setting?

Round 2

Reviewer 1 Report

The authors have improved and restructured the manuscript. However, they could improve the English language.

Author Response

Dear reviewer,

Thank you for your reply. We took another look at the article and tried our best to exclude all remaining mistakes. Also, we tried to enhance our English level a bit.

We appreciate all feedback we obtained from you.

Best regards

Vojtěch Jašek